# A natural history study to track brain and spinal cord changes in individuals with Friedreich's ataxia: TRACK-FA study protocol

**Nellie Georgiou-Karistianis**[1]*, Louise A. Corben[2,3], Kathrin Reetz[4,5], Isaac M. Adanyeguh[6], Manuela Corti[7], Dinesh K. Deelchand[6], Martin B. Delatycki[1,2,3], Imis Dogan[4,5], Rebecca Evans[8], Jennifer Farmer[9], Marcondes C. França[10], William Gaetz[11], Ian H. Harding[12], Karen S. Harris[1], Steven Hersch[13], Richard Joules[14], **James J. Joers**[6], Michelle L. Krishnan[15], Michelle Lax[14], Eric F. Lock[16], David Lynch[17], Thomas Mareci[18], Sahan Muthuhetti Gamage[1], Massimo Pandolfo[19], Marina Papoutsi[14], Thiago J. R. Rezende[10], Timothy P. L. Roberts[11], **Jens T. Rosenberg**[20], Sandro Romanzetti[4,5], Jörg B. Schulz[4,5], Traci Schilling[21], Adam J. Schwarz[8], Sub Subramony[20], Bert Yao[21], Stephen Zicha[8], Christophe Lenglet[6], Pierre-Gilles Henry[6]

1 School of Psychological Sciences, The Turner Institute for Brain and Mental Health, Monash University, Clayton, Victoria, Australia, 2 Bruce Lefroy Centre for Genetic Health Research, Murdoch Children's Research Institute, Parkville, Victoria, Australia, 3 Department of Paediatrics, University of Melbourne, Parkville, Victoria, Australia, 4 Department of Neurology, RWTH Aachen University, Aachen, Germany, 5 JARA-BRAIN Institute Molecular Neuroscience and Neuroimaging, Forschungszentrum Jülich GmbH and RWTH Aachen University, Aachen, Germany, 6 Center for Magnetic Resonance Research and Department of Radiology, University of Minnesota, Minneapolis, Minnesota, United States of America, 7 Powell Gene Therapy Centre, University of Florida, Gainesville, Florida, United States of America, 8 Takeda Pharmaceutical Company Ltd, Cambridge, Massachusetts, United States of America, 9 Friedreich's Ataxia Research Alliance (FARA), Downingtown, Pennsylvania, United States of America, 10 Department of Neurology, University of Campinas, Campinas, Sao Paulo, Brazil, 11 Department of Radiology, Lurie Family Foundations MEG Imaging Center, Children's Hospital of Philadelphia, Philadelphia, Pennsylvania, United States of America, 12 Department of Neuroscience, Central Clinical School, Monash University, Melbourne, Australia, 13 Neurology Business Group, Eisai Inc., Nutley, New Jersey, United States of America, 14 IXICO plc, London, England, 15 Translational Medicine, Novartis Institutes for Biomedical Research, Cambridge, MA, United States of America, 16 Division of Biostatistics, School of Public Health, University of Minnesota, Minneapolis, MN, United States of America, 17 Department of Neurology, Children's Hospital of Philadelphia, Philadelphia, Pennsylvania, United States of America, 18 Department of Biochemistry and Molecular Biology, University of Florida, Gainesville, FL, United States of America, 19 Department of Neurology and Neurosurgery, McGill University, Montreal, Canada, 20 McKnight Brain Institute, Department of Neurology, University of Florida, Gainesville, Florida, United States of America, 21 PTC Therapeutics, Inc, South Plainfield, New Jersey, United States of America

* nellie.georgiou-karistianis@monash.edu

## Abstract

### Introduction

Drug development for neurodegenerative diseases such as Friedreich's ataxia (FRDA) is limited by a lack of validated, sensitive biomarkers of pharmacodynamic response in affected tissue and disease progression. Studies employing neuroimaging measures to track FRDA have thus far been limited by their small sample sizes and limited follow up. TRACK-FA, a longitudinal, multi-site, and multi-modal neuroimaging natural history study, aims to address these shortcomings by enabling better understanding of underlying pathology and identifying sensitive, clinical trial ready, neuroimaging biomarkers for FRDA.

**Data Availability Statement:** This article is a study protocol and does not report data; therefore, the data availability policy is not applicable.

**Funding:** This study is funded by grants from the Friedreich's Ataxia Research Alliance (FARA) to each of the academic sites and IXICO plc with financial support from Takeda Pharmaceuticals Company Ltd, Novartis Gene Therapies, IXICO plc and PTC Therapeutics. The Friedreich Ataxia Research Alliance does not use grant numbers. The sponsor of the study is Monash University The study sites and the site principle Investigators (main authors who received funding) are detailed below. Monash University (N.G.K) University of Minnesota (P.G.H) Children's Hospital of Philadelphia (D.L) University of Florida (S.S) RWTH Aachen University (K.R) University of Campinas (M.C.F) McGill University (M.P) Funding bodies Friedreich's Ataxia Research Alliance (FARA): https://www.curefa.org/ IXICO plc: https://ixico.com/ Novartis Gene Therapies: https://www.novartis.com/about/innovative-medicines/novartis-pharmaceuticals Takeda Pharmaceuticals company Ltd: https://www.takeda.com/en-au/ PTC Therapeutics:https://www.ptcbio.com/ FARA plays an ongoing role in study oversight, study design, decision to publish and author J.F who is J.F. is employed by the Friedreich's Ataxia Research Alliance (FARA) played a role in the preparation of the manuscript. Takeda Pharmaceuticals Company Ltd plays an ongoing role in study oversight, study design, decision to publish and authors R.E. and S. Z are employees of Takeda Pharmaceuticals Company Ltd and played a role in preparation of the manuscript. Additionally former employee of Takeda Pharmaceuticals A.S. also contributed to the preparation of the manuscript. PTC Therapeutics plays an ongoing role in study oversight, study design, decision to publish and authors T.S. and B.Y. who are employees of PTC Therapeutics contributed to the preparation of the manuscript. Novartis Gene Therapies plays an ongoing role in study oversight, study design, decision to publish and authors M.L.K who holds shares in Novartis Gene Therapies as indicated in the conflicts on interest played a role in the preparation of the manuscript. IXICO plc plays an ongoing role in study oversight, study design, decision to publish will perform independent quality control and data analysis of brain anatomical imaging and brain diffusion imaging data for one third of TRACK-FA participants. Authors M.L, R.J and M.P are employees of IXICO and contributed to the preparation of the manuscript.

## Methods

200 individuals with FRDA and 104 control participants will be recruited across seven international study sites. Inclusion criteria for participants with genetically confirmed FRDA involves, age of disease onset $\leq$ 25 years, Friedreich's Ataxia Rating Scale (FARS) functional staging score of $\leq$ 5, and a total modified FARS (mFARS) score of $\leq$ 65 upon enrolment. The control cohort is matched to the FRDA cohort for age, sex, handedness, and years of education. Participants will be evaluated at three study visits over two years. Each visit comprises of a harmonized multimodal Magnetic Resonance Imaging (MRI) and Spectroscopy (MRS) scan of the brain and spinal cord; clinical, cognitive, mood and speech assessments and collection of a blood sample. Primary outcome measures, informed by previous neuroimaging studies, include measures of: spinal cord and brain morphometry, spinal cord and brain microstructure (measured using diffusion MRI), brain iron accumulation (using Quantitative Susceptibility Mapping) and spinal cord biochemistry (using MRS). Secondary and exploratory outcome measures include clinical, cognitive assessments and blood biomarkers.

## Discussion

Prioritising immediate areas of need, TRACK-FA aims to deliver a set of sensitive, clinical trial-ready neuroimaging biomarkers to accelerate drug discovery efforts and better understand disease trajectory. Once validated, these potential pharmacodynamic biomarkers can be used to measure the efficacy of new therapeutics in forestalling disease progression.

## Clinical trial registration

ClinicalTrails.gov Identifier: NCT04349514.

## Introduction

Friedreich's ataxia (FRDA) is the most common inherited ataxia, with an estimated incidence in of 1 in 29,000 and a carrier frequency of 1 in 85 individuals of Caucasian background [1, 2]. Neurological features include progressive gait and limb ataxia, absent lower limb reflexes, extensor plantar responses, dysarthria and loss of position and vibration sense [3]. Non-neurological features include cardiomyopathy, visual and hearing impairments, diabetes mellitus, glucose intolerance and skeletal deformities [4]. Onset of FRDA occurs, on average, at 10–15 years, but may be delayed to the middle or later years of life.

The genetic cause of FRDA is mutation of the *FXN* gene, primarily in the form of homozygous GAA trinucleotide repeat expansions [5, 6]. The age of disease onset and associated severity of some symptoms is inversely correlated with the size of the GAA expansion on the smaller of the *FXN* alleles (GAA1) [7]. This genetic mutation results in reduced expression of frataxin, an essential protein in the Iron-Sulfur (FE-S) cluster complex, which disrupts iron homeostasis [8, 9]. Frataxin is highly expressed in the dorsal root ganglia (DRG), the spinal cord, cerebellar dentate nuclei, cerebral cortex, pancreas, heart, liver and skeletal muscle [5], reflecting the tissues most impacted in FRDA.

The neuropathology underlying FRDA is multifaceted and complex. Hypoplasia of large and small neurons of the DRG, as well as effects on the peripheral nerves and spinal cord are

**Competing interests:** I have read the journal's policy and the authors of this manuscript have the following competing interests: R.E. and S.Z. are employed by Takeda Pharmaceutical Company Ltd with both authors receiving salary and S.Z. holds stocks in the company. A.S. was employed by Takeda Pharmaceutical Company Ltd at the time of his contribution to the TRACK-FA project. Takeda Pharmaceutical Company Ltd remains committed to FRDA research and will help develop translational tools to monitor patient disease and share with the FRDA community. T.S. and B.Y. are both employees of PTC Therapeutics. D.L. is a grant recipient from the National Institute of Health (NIH), Muscular Dystrophy Association (MDA), Friedreich's Ataxia Research Alliance (FARA), Reata Pharmaceuticals Inc, Retrotope Inc, Voyager Therapeutics, Novartis Gene Therapies, Audentes Therapeutics (Astellas Gene Therapies) and Minoryx Therapeutics S.L. T.P.L.R. has equity interest in PRISM Clinical Imaging and Proteus Neurodynamics and consulting/advisory board engagement with CTF MEG International Services LP, Ricoh Company Ltd, Spago Nanomedical AB, Avexis (Novartis Gene Therapies) and Acadia Pharmaceuticals Inc. P.G.H. is a grant recipient from the Friedreich's Ataxia Research Alliance (FARA), GoFAR, Ataxia UK, the Bob Allison Ataxia Research Centre, and the National Institute of Health (NIH). CMRR is supported by NIH grants P41EB027061 and P30NS076408. P.G.H. reports grants from Minoryx Therapeutics for activities outside this study. M.P. and R.J. and M.L. are employed by IXICO plc, ML is a shareholder for IXICO plc. M.C.F. is a grant recipient from PTC Therapeutics and has taken part in advisory board for PTC Therapeutics and Avexis (Novartis Gene Therapies). T.R. is a grant recipient from the Friedreich's Ataxia Research Alliance (FARA). C.L. is a research grant recipient from the Friedreich's Ataxia Research Alliance (FARA), GoFAR, Ataxia UK, the Bob Allison Ataxia Research Center, and National Institute of Health (NIH) grants P41. EB027061 and P30 NS076408. C.L. reports research grants from Minoryx Therapeutics and Biogen Inc. for activities outside this study. S.S. is a broad member of the Research Advisory Board for National Ataxia Foundation (USA), a research grant recipient from the Friedreich's Ataxia Research Alliance (FARA), Wyck Foundation, National Ataxia Foundation, Muscular Dystrophy Association (MDA), National Institute of Health (NIH), FDA and receives industry support from Reata Pharmaceutical Inc, Retrotope Inc, PTC Therapeutics, Biohaven Pharmaceuticals, Avidity Biosciences Inc, and Strides Pharma Science Limited. M.L.K. holds shares in Novartis Gene

observed and suggests neurodevelopmental origin [10, 11]. Over time, atrophy of the dentate nucleus becomes apparent, significantly contributing to the neurological phenotype [10, 12, 13]. Abnormalities in cerebral structures [14–16], functions [17–20] and both intra-cerebral and cerebello-cerebral connectivity [17, 21–25] have also been reported in individuals with FRDA.

Currently, there are no approved disease-modifying treatments for FRDA. Trials to investigate new drugs typically employ broad measures of clinical and functional progression, which require large sample sizes and long observational periods to detect therapeutic efficacy [26–29]. Useful biomarkers that can more rapidly and sensitively identify tissue-specific effects, thus improving clinical trial efficiency, are therefore urgently required to enable early decision making and to support potential regulatory considerations.

There have been significant efforts over the last few years to build neuroimaging studies in FRDA, in order to track the trajectory of disease progression at the brain and spinal cord levels. This is evidenced in longitudinal studies such as IMAGE-FRDA [30, 31], which has provided crucial information on volumetric and microstructural changes in the cerebellar peduncles and brainstem [19, 20, 24]. Another longitudinal study provided information on volumetric, microstructural, and neurochemical changes in the spinal cord [32]. The ENIGMA-Ataxia project [33], a large cross sectional multi-site retrospective study, further helped identify regions altered in the brain at different stages of the disease. Collaborative work such as the European Friedreich's Ataxia Consortium for Translational studies (EFACTS) [26–28], and the Friedreich's Ataxia Clinical Outcome Measures (FACOMS) [34, 35] helped evaluate longitudinal changes in clinical assessment tools and develop assays to measure blood frataxin. Finally, the Human Connectome Project [36], and the standardised spinal cord imaging protocol (Montreal Protocol) [37] spearheaded the development of new, more efficient and standardised MRI protocols and analysis procedures.

This culmination of information and techniques from a range of existing studies has highlighted that neuroimaging measures of the brain and spinal cord are promising biomarkers of pharmacodynamic response to potential interventions within target tissues, and for tracking disease progression in FRDA [38]. Previous single site studies have shown associations between clinical severity and imaging measures such as the cross-sectional area of the cervical spinal cord [39], volume and iron content of the dentate nucleus [13, 40], microstructural integrity of the brainstem white matter [15, 24]. Furthermore, longitudinal measurements have shown that spinal cord morphometry (cross-sectional area), spinal cord fractional anisotropy, and spinal cord spectroscopy (ratio tNAA/mIns) are sensitive to disease progression in an early-stage cohort [32]. Importantly, longitudinal analysis has shown that particular neuroimaging measures may be more sensitive to FRDA progression at different stages of disease [13]. For example, the rate of change in iron concentration in the dentate nucleus accelerates with disease progression, whilst the rate of atrophy in this region is at its highest early in the disease course, plateauing over time [13]. Others have shown that clinical severity, genotype, age, and age at symptom onset influence the underlying neuropathology [15, 16, 24, 41]. Understanding the relationship between these parameters will therefore be crucial to establishing which neuroimaging biomarkers are most sensitive to disease progression, whether a biomarker is able to detect early onset of disease related symptoms and which biomarkers are most sensitive to potential treatments, within specific cohorts.

These previous studies have identified brain and spinal cord regions that may be compromised in FRDA, providing possible targets for interventional clinical trials, and biomarkers of disease progression. However, relatively small sample sizes, limited longitudinal follow up, and differing methodology at individual clinical research centres have limited the ability to achieve the necessary statistical power to detect significant changes, and to perform targeted sub-

Therapies. K.R. has received grants from the German Federal Ministry of Education and Research (BMBF 01GQ1402, 01DN18022), the German Research Foundation (IRTG 2150, ZUK32/1), Alzheimer Forschung Initiative e.V. (AFI 13812, NL-18002CB) and honoraria for presentations or advisory boards from Biogen and Roche. J.F. is employed by the Friedreich's Ataxia Research Alliance (FARA) and receives a salary from this institution. L.C. is a research grant recipient from the Friedreich Ataxia Research Alliance (FARA), Ataxia UK, Medical Research Future Fund and is funded by a Medical Research Futures Fund Next Generation Career Development Fellowship.

group analyses to better understand the neuropathology within specific cohorts. Furthermore, expanded longitudinal investigations are essential to distinguish developmental from degenerative changes, to identify the measurements most sensitive to clinical changes at different points in the disease trajectory and to more accurately relate imaging alterations with particular aspects of the complex clinical syndrome.

Disease trajectory mapping can also determine temporal changes that occur in FRDA and specific targets for future gene and cell therapy [42, 43], complementing the current gene therapy approaches treating cardiac complications in FRDA [43–45]. Furthermore, imaging biomarkers can be used as targets for potential therapeutics, such as those that augment mitochondrial function (Idebenone) [46, 47], or those that aim to increase frataxin levels (Erythropoietin) [48, 49], aiding in the evaluation of their efficacy, pharmacokinetic and pharmacodynamic properties [50].

In building on from this body of work, and understanding previous limitations, we conceived TRACK-FA: a longitudinal, multi-site and multi-modal neuroimaging natural history study designed to refine and validate potential neuroimaging biomarkers of FRDA progression. TRACK-FA will combine multiple imaging modalities–structural MRI, diffusion MRI, quantitative susceptibility mapping (QSM), and magnetic resonance spectroscopy (MRS)–to obtain complementary structural, microstructural and biochemical information on the brain and spinal cord in FRDA compared to control cohorts over 24 months. This combination of modalities will provide a more detailed picture of the underlying neuropathology in FRDA. Furthermore, its multi-site nature will facilitate the sample sizes needed to perform statistically powerful subgroup analysis.

TRACK-FA brings together global expertise in FRDA across academic, industry and advocacy groups to undertake the world-first prospective multi-site natural history study specifically to assess neurobiological changes underlying FRDA. TRACK-FA is scientifically governed by the TRACK-FA Neuroimaging Consortium and aims to accelerate the discovery of sensitive neuroimaging biomarkers of disease severity and progression to improve clinical trial efficiency.

## Methods and analysis

### Study aims

TRACK-FA aims to assess the potential value of imaging biomarkers and provide a scientific basis for instituting them in clinical trials for FRDA.

### Objectives

**Objective 1:** To determine the longitudinal sensitivity of a range of candidate neuroimaging biomarkers in the brain and spinal cord and identify the biomarker(s) most sensitive to change over time in individuals with FRDA compared to control participants.

**Objective 2:** To determine differences in candidate neuroimaging biomarkers and their time of change between children and adults with FRDA, and matched controls.

**Objective 3:** To determine change over time in clinical severity, cognitive function, mood, speech, and fluid markers such as blood Neurofilament Light Chain (NFL) and frataxin in individuals with FRDA compared to controls.

**Objective 4:** To determine the relationship between imaging biomarkers and clinical, cognitive, mood, speech and blood markers of disease severity, including age of onset, GAA repeat size, disease symptoms and duration, disease severity, frataxin levels and NFL in individuals with FRDA.

## Study setting

This study comprises multiple international sites including: Monash University and Murdoch Children's Research Institute (Melbourne, Australia), Rheinisch-Westfälische Technische Hochschule Aachen [(RWTH Aachen University), Aachen, Germany], University of Minnesota (Minnesota, USA), University of Florida (Florida, USA), University of Campinas (Campinas, Brazil), McGill University (Montreal, Canada) and the Children's Hospital of Philadelphia (CHOP) (Philadelphia, USA).

## Ethics statement

This study will be conducted in accordance with the Declaration of Helsinki (revised in 2013 by Fortaleza) and Ethical Guidelines for Medical Research Involving Human Subjects (partially revised on February 28, 2017).

This study has obtained approval from the institutional review boards (IRBs) at each of the study sites. The reference numbers for each study site are provided below.

Monash Health Human Research Ethics Committee (HREC): RES-20-0000-139A

Children's Hospital of Philadelphia: IRB 20–017611

University of Minnesota: IRB STUDY00009047

University of Florida: IRB202000399

RWTH Aachen University: EK195/20

University of Campinas (CAAE NO): 83241318.3.1001.5404

McGill University (under review)

Potential participants will be provided with information about the study via an IRB-approved letter and information sheet. Research personnel will explain all study procedures to potential participants. Participants will be enrolled into the study upon obtaining written informed consent and meeting the inclusion and exclusion criteria (see below). Opt-in and opt-out systems are in place for all participants. All participant data will be de-identified upon enrolment into the study using a unique participant identifier. Each site will be required to ensure that participants are consented in such a way that allows the sharing of de-identified data (as outlined below).

## Data sharing

It is recognised that this project will generate data that are of interest to the FRDA academic and bio-pharmaceutical/drug development community. All de-identified data will be made available to consortium members after being generated, and then later to additional third parties at the completion of the study upon request, with approval from the TRACK-FA Steering Committee. All parties will have full access to the de-identified raw data throughout the study and interim reporting and final reports will be provided throughout the study in accordance with the milestones and deliverables.

All academic partners will provide access to their data to the TRACK-FA Neuroimaging Consortium members and the study funders, as well as the right to create derivative works of the research data. Up to date information regarding data sharing is available at the TRACK-FA website: Track-FA Study (monash.edu)

## Study design

The TRACK-FA study will involve the participation of children and adults with genetically confirmed FRDA and control participants. Participants will be allocated to one of four groups based on their age: $\geq 18$ years, 11–17 years, 8–10 years, and 5–7 years. Control participants

will be matched at the cohort level to the FRDA participants for age, sex, handedness, and years of education. A recruitment ratio of 1:1 (FRDA: control) will be used for those aged 5–10 years and a 2:1 ratio (FRDA: control) will be used for participants ≥ 11 years of age.

We aim to recruit a total of 200 individuals with FRDA and 104 control participants across seven testing sites. This number was informed by a power analysis (see below) and necessitated a multi-site collaborative network to achieve this goal. All participants will then be assessed at three visits [Visit 1 (V1), Visit 2 (V2), Visit 3 (V3)] with identical assessments; each visit will take place approximately 12 months apart and no more than two months before or after the 12-month interval.

**Inclusion and exclusion criteria.** Participants with FRDA are eligible for TRACK-FA if they meet the inclusion and exclusion criteria as outlined in Table 1. Due to the nature of this criteria, this study is only targeting those individuals with FRDA that are largely ambulant and in the first 25 years of disease duration.

Control participants must meet the following criteria: age > 5 years and be able to provide written informed consent. Control participants will be excluded if they have any MR contraindications, are pregnant, or present with any diagnosed medical and neurological condition which may interfere with the conduction of the study.

**Stratification of participants.** The Friedreich's Ataxia Rating Scale (FARS) functional staging for ataxia scale [51], will be used to stratify participants with FRDA into disease stage subgroups as outlined in Table 2. At the time of enrolment, we aim to have 55% of participants in FARS functional stages 0–2, 35% in the functional stages 3–4 and 10% in functional stage 5. Individuals in stage 6 will not be enrolled. This will create a cohort largely representative of those earlier in the disease process. Those earlier in the course of disease, particularly those who are still ambulant, progress at a much faster rate [27, 52, 53], hence detecting progression through proposed biomarkers in this cohort may be more efficient.

**Study visits.** Each study visit will include a clinical assessment (most items applicable to individuals with FRDA only), cognitive and mood assessments, collection of demographic and medical information, MRI of the brain and spinal cord, and collection of a blood sample. The schedule followed by each participant is dependent on their age group (Fig 1).

*Collection of medical and demographic information.* Medical history specific to FRDA will be taken in the enrolment session for all participants with FRDA. The following information will be collected: the length of *FXN* GAA repeat expansions on the smaller (GAA1) and larger

**Table 1. Inclusion and exclusion criteria used for the enrolment of FA participants into the TRACK-FA study.**

| Inclusion criteria below | Exclusion criteria below |
|---|---|
| Age ≥ 5 years | Acute or ongoing medical or other conditions not attributed to FRDA |
| Biallelic GAA repeat length > 55 in intron 1 of *FXN* and/or GAA repeat length > 55 in intron 1 of *FXN* in one allele and another type of mutation that is inferred to cause loss of function in the second *FXN* allele | Other neurological condition apart from FRDA |
| Age of disease onset ≤ 25 years | MR contraindications (e.g., pacemaker or other metallic surgical implants) |
| Disease duration ≤ 25 years | Presence of metallic dental braces (creates unwanted artifacts) |
| Written informed consent provided | Pregnancy |
| *FARS functional stating score of ≤ 5 and a total mFARS score of ≤65 on enrolment | |

*FARS, Friedreich's Ataxia Rating Scale; mFARS, Modified Friedreich's' Ataxia Rating Scale.

**Table 2. The FARS Functional Staging for Ataxia scale describing the clinical presentation at each stage.**

| Friedreich's Ataxia Rating Scale Functional Staging for Ataxia Scale | |
| --- | --- |
| **Functional staging** | **Clinical presentation** |
| Stage 0 | Normal |
| Stage 1 | Minimal signs detected by physician during screening. Can run or jump without loss of balance. No disability |
| Stage 2 | Symptoms present, recognized by patient, but still mild. Cannot run or jump without losing balance. The patient is physically capable of leading an independent life, but daily activities may be somewhat restricted. Minimal disability |
| Stage 3 | Symptoms are overt and significant. Requires regular or periodic holding onto wall/furniture or use of a cane for stability and walking. Mild disability. (Note: many patients postpone obtaining a cane by avoiding open spaces and walking with the aid of walls/ people etc. These patients are graded as stage 3.0) |
| Stage 4 | Walking requires a walker, Canadian crutches or two canes and or other aids such as walking dogs. Can perform several activities of daily living. Moderate disability |
| Stage 5 | Confined but can navigate a wheelchair. Can perform some activities of daily living that do not require standing or walking. Severe disability |
| Stage 6 | Confined to wheelchair or bed with total dependency for all activities of daily living. Total disability |

(GAA2) allele, the type of mutation detected, age of symptom onset including first symptom, disease duration, presence of diabetes and medical information pertaining to nervous system disorders (apart from FRDA), heart conditions, vision impairments, skeletal abnormalities (scoliosis and foot deformities), hearing conditions, bowel and bladder conditions, presence of inflammatory diseases, details on concurrent trial participation and COVID-19 status.

The following information will be collected for all participants: height and weight (used to calculate body mass index (BMI), and medications (prescription/non-prescription).

*MRI scanning to assess primary outcomes.* All participants will undergo brain and (cervical) spinal cord imaging procedures lying supine in a three tesla (3T) MRI scanner (Siemens Skyra,

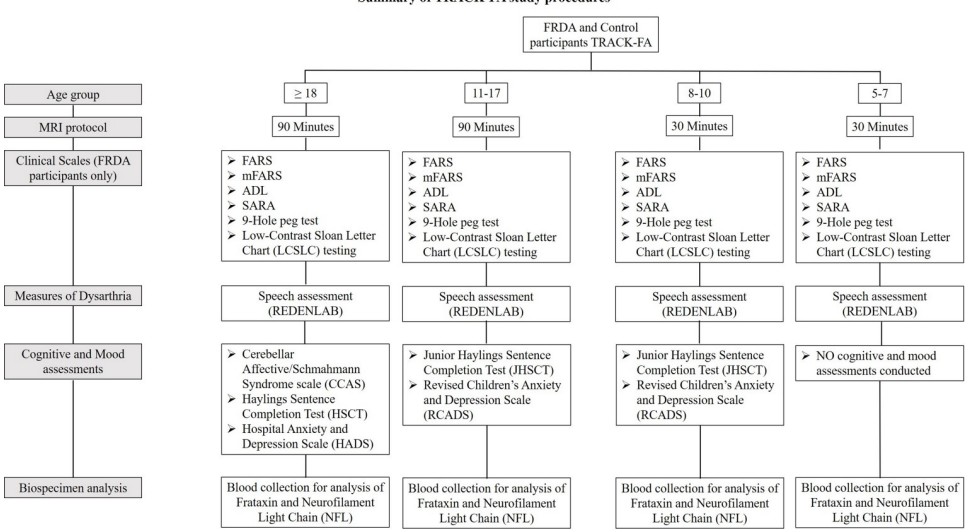

**Fig 1. Summary of TRACK-FA study procedures according to participant age.** The testing schedule varies according to age. All participants will also undergo a speech assessment, whilst all FRDA participants will undergo additional clinical testing.

Siemens Prisma or Philips Ingenia). Significant effort has been made to ensure harmonisation of MR protocols across all the sites with vigorous testing of the imaging sequences prior to the commencement of the study. The inclusion of QSM, diffusion MRI and MRS in the procedure allows for the quantification of iron accumulation, microstructural and neuro-chemical profile of areas pertinent to disease pathology [39], potentially providing further insights into neuronal damage and gliosis associated with the neurodegenerative nature of this disease [32].

Depending on age, one of two MRI protocols will be used: the first protocol, approximately 90 minutes long, will be used in all participants aged >10 years. The second, approximately 30 minutes long and designed to be less taxing for younger children, will be used in the 5–10 years age group. The young child protocol only includes anatomical scans (brain and spinal cord), but no diffusion MRI, brain QSM, or spinal cord MRS. Brain T1, T2 and diffusion acquisitions are based on those used in the Human Connectome Project [36]. Brain QSM acquisition was optimised to provide excellent image quality and contrast with 0.9 isotropic resolution. The spinal cord T2, diffusion and MEDIC acquisitions are based on a recently published spinal cord protocol [37]. The spinal cord MRS is based on a semi-LASER sequence developed at the University of Minnesota [54], with addition of metabolic cycling. The specific MR sequences, acquisition time, and key parameters for each sequence are outlined below in Table 3. These sequences will be used to acquire the outcome measures provided in Table 4.

*Clinical measures to assess secondary outcomes.* Individuals with FRDA will undergo assessments using neurologic scales and functional measures outlined in Table 5. Additionally, all participants enrolled will undergo speech assessments. The speech assessment is conducted using the REDENLAB software [58], consisting of an adult battery (for those aged ≥ 18 years) and a paediatric battery (aged 5–17 years).

*Cognitive and mood assessments to assess secondary outcomes.* Individuals with FRDA and control participants will undergo a range of cognitive and mood assessments for the TRACK-FA study. These assessments have been outlined in Table 6. Paediatric participants aged < 8 years will not be required to undergo measures of cognition or mood.

**Table 3. TRACK-FA protocol used to obtain cervical spinal cord and brain MRI data.**

| Data | Sequence | Resolution | Main parameters | Acq. time |
|---|---|---|---|---|
| Spine T2 | 3D SPACE | 0.8 mm iso | TE 120 ms, TR 1500ms, iPAT 3 | 4'02" |
| Spine MRS battery (includes power calibration and shimming) | semiLASER (CMRR) | 8 x 6 x 33 mm | TE 30 ms, TR 2500 ms, NA 160 | 8'10"" |
| Spine diffusion | 2D SE-EPI with ZOOMit | 0.9 x 0.9 x 5mm | TE 61 ms (Skyra 92ms), TR 620 ms (Skyra 680 ms), 5 concatenations, 30 dirs, b = 800, 5 b0 On Skyra, NT = 2 for diffusion weighted volumes (b = 800) | 4'(AP) + 1' (PA) (b0 only) (with pulse trigger) On Skyra these times are nearly doubled) |
| Spine MEDIC | 2D multi-echo | 0.5 x 0.5 x 5mm | TE 14 ms, TR 600 ms, NA 2, 3 echoes, iPAT 2 | 4'45" |
| Brain T1 | 3D MPRAGE | 0.8 mm iso | TE 2.2ms, TR 2400 ms, TI 1000 ms, iPAT 2 | 6'38" |
| Brain T2 | 3D SPACE | 0.8 mm iso | TE 563ms, TR 3200 ms, iPAT 2 | 5'57" |
| Brain diffusion | Multiband EPI (CMRR) | 1.5 mm iso (Skyra 1.7 mm iso) | TE 89ms (Skyra 102 ms), TR 3230 ms (Skyra 3366 ms), MB 4, 197 dirs, 2 shells, b = 1500/3000, 6 b0 | 5'37" (AP) + 5'37" (PA) |
| | | | | Skyra 5'52" (AP) + 5'52" (PA) |
| Brain QSM | 3D GRE | 0.9 mm iso | TE1/TE2/TE3/TE4 = 3.7/9.7/15.8/21.9 ms, TR 27ms, iPAT 2 | 7'22" |

* For participants 10 years old and younger, only anatomical data (highlighted in grey) are collected.

** The times above are data acquisition times for each sequence and do not include time for adjustments (e.g., scout images, shimming, power calibration) and ROI planning.

**Table 4. The brain regions, modalities, and outcome measures for the TRACK-FA MRI protocol.**

| Region | Modality | Outcome measure |
|---|---|---|
| Brain | Structural MRI (Brain morphometry) | Total brain volume |
| | | Total white matter volume |
| | | Total grey matter volume |
| Brain | Structural MRI (Cerebellum morphometry) | Total cerebellar volume |
| | | Superior cerebellar peduncles volume |
| Brain | Quantitative susceptibility mapping (cerebellum only) | Iron concentration in the dentate nuclei* |
| | | Volume of the dentate nuclei* |
| Brain | Diffusion MRI | Superior cerebellar peduncle fractional anisotropy* |
| | | Superior cerebellar peduncle mean diffusivity* |
| | | Superior cerebellar peduncle radial diffusivity* |
| | | Superior cerebellar peduncle axial diffusivity* |
| Spinal cord | Structural MRI (morphometry) (Values extracted from the spine T2 image) | Cross sectional area of the cervical spinal cord, averaged over two C2-C3 |
| Spinal cord | Diffusion MRI | Total White matter changes in fractional anisotropy*, mean diffusivity*, radial diffusivity*, axial diffusivity* |
| Spinal cord | Magnetic resonance spectroscopy (Voxel positioned over C4-C5) | Total N-acetyl-aspartate (tNAA)/myo-inositol (mIns) ratio* |

(*) only undertaken for participants > 10 years of age.

*Blood collection to assess exploratory outcomes.* A blood sample will be collected from each participant by a certified blood collection professional. The samples will be analysed for two proteins, frataxin and NFL, directly from whole blood, and from serum samples. At all visits, blood will be collected into serum separator tubes (SSTs), coagulated at ambient room

**Table 5. Neurologic and functional measures completed by TRACK-FA participants.**

| Assessment | Description |
|---|---|
| FARS Staging for Ataxia Scale | This assessment indicates the impact of FRDA on function. |
| mFARS neurological exam (FARS subscale) | The mFARS comprises four subscales (bulbar, upper limb coordination, lower limb coordination, and upright stability) and has a maximum score of 93 [34] |
| FARS Activities of Daily Living (ADL) | The ADL, assessed in a structured guided interview setting, is a component of the FARS and aims to quantify essential and routine aspects of self-care, The ADL comprises 9 questions and has a maximum score of 36 [51] |
| Scale for Assessment and Rating of Ataxia (SARA) | The SARA is a semi-quantitative assessment of ataxia, measuring ataxia of upper limb, lower limb, gait, balance and speech. It has eight items, with the total score ranging from zero (no ataxia) to 40 (severe ataxia) [55] |
| 9- Hole-Peg test (9HPT) | This test examines finger dexterity and involves placing and removing nine pegs in a pegboard in the quickest possible time. The 9HPT has high intra- and inter-rater reliability and is the most commonly used measure of upper limb function in FRDA [56] |
| Low-Contrast Sloan Letter Chart (LCSLC) testing | Contrast letter acuity testing for vision will be performed using the low-contrast Sloan letter charts (LCSLC–Precision Vision, La Salle, IL) [57]. Binocular LCSLC measurements will be obtained at 2 meters for each of three contrast levels (100%—equivalent to high-contrast visual acuity, 2.5%, and 1.25% contrast levels) |
| *Measures of Dysarthria | Dysarthria is measured using speech testing. The specific speech testing software used is the REDENLAB software [58] consisting of an adult and paediatric battery |

*Clinical assessments undertaken by both individuals with FRDA and control participants.

**Table 6. The cognitive and mood assessments completed by TRACK-FA participants.**

| Assessment | Description |
|---|---|
| Cerebellar Cognitive Affective/Schmahmann Syndrome Scale (CCAS) | The CCAS is a 10-item scale assessing the executive, visual spatial and linguistic components of cognitive control and affect [59]. This assessment will be only completed by those in the ≥18 age group |
| Hayling Sentence Completion Test (HSCT)/ Junior HSCT | The HSCT will be completed by those in the ≥ 18 age group, while the Junior HSCT [60] will be administered to the 11–17 and 8–10 age groups. The HSCT is a measure of response initiation and suppression [60–62], and has demonstrated sensitivity to detect cognitive deficits in FRDA [63] |
| The Hospital Anxiety and Depression Scale (HADS)/ Revised Children's Anxiety and Depression Scale (RCADS) | The anxiety and depression subscales are valid measures for severity of emotional disorder [64] with specific insights into variables such as: separation anxiety disorder, social phobia, generalized anxiety disorder, panic disorder, obsessive compulsive disorder, and major depressive disorder [65]. The Revised Children's Anxiety and Depression Scale (RCADS) [66] will be completed by those in the 11–17 and 8–10 age groups |

temperature for 30 minutes and centrifuged. The samples will be shipped on dry ice at approximately six-monthly intervals to laboratories equipped to carry out frataxin and NFL analysis.

For those participants who do not have a record of GAA repeat sizing, extra blood will be drawn to conduct this analysis.

**Adverse events.**    All adverse events will be captured by study sites and will be categorised according to the Standard Protocol Items: Recommendations for Interventional Trials (SPIRIT) guidelines [67]. Briefly, a Serious Adverse Event (SAE) is defined as an untoward or unexpected medical occurrence that results in death, is life-threatening, requires hospitalisation, results in persistent or significant disability or incapacity, or is a congenital anomaly or birth defect. Suspected Unexpected Serious Adverse Reactions (SUSAR) to medicinal products and other types of Adverse Events, including Non-Serious Adverse Events, Serious Adverse Events not related to TRACK-FA procedures, Non-Serious Adverse Reactions to medicinal products, and Serious Adverse Reactions to medicinal products will also be recorded. All Adverse Events, regardless of seriousness, and as related to TRACK-FA procedures, will be reported into a central repository, and reviewed at each quarterly meeting by the TRACK-FA Steering Committee. Sites will be obliged to report all Serious Adverse Events related to TRACK-FA procedures to the TRACK-FA Steering Committee within 24 hours via the PI/Safety Officer.

## Data analysis and statistical methods

**Sample size calculation.**    The sample size required for this study was determined to satisfy a specified level of statistical power to detect a change in each measured parameter over the 24-month period. A minimum effect size of Cohen's $d = 0.3$ was selected. The corresponding minimum number of participants (n) to detect such a change ($p$-value<0.05) with power greater than 0.80 was $n = 70$. We then doubled "n" to investigate changes in two sub cohorts: early stage and later stage yielding an $n = 140$. An important caveat is that the rate of change will be likely different in each sub-cohort, hence the power will be different in each sub-cohort. The final step was to add an additional 25% to the "$n$" to account for a possible 20% participant attrition over 24 months. An estimated recruitment size of $n = 200$ was selected. The TRACK-FA team includes experienced biostatisticians who will continue to advise on all statistical analyses throughout the study.

## Statistical analysis plan (SAP)

*MRI data analysis.* The MRI protocol includes the accumulation of multiple images using various MRI modalities as described in Table 3. The modalities are: brain structural MRI (T1 and T2 weighted), cerebellum structural MRI (using the same brain images but cerebellum-specific processing pipelines), brain diffusion MRI, brain QSM, spinal cord structural MRI, spinal cord diffusion MRI and spinal cord MRS. For each of these modalities, all data from all seven sites will be processed by a single site to ensure consistent data processing.

Brain morphometry (structural imaging) will be performed using FreeSurfer [68, 69] to determine brain total volume, total white matter volume and total grey matter volume. Cerebellum morphometry, performed using ACAPULCO [70], to determine the total cerebellar volume while SUIT [71] will be used to calculate the volume of left and right cerebellar peduncles. Brain DTI data will be analysed using FMRIB Software Library (FSL) [69, 72] to determine the fractional anisotropy (FAn) and mean, axial and radial diffusivity (MD, AD, RD) of the left and right SCP. Brain QSM data will be reconstructed using the JHU/KKI QSM Toolbox v3.0 from the MATLAB suite [73, 74], and the left and right dentate nucleus will be segmented manually [13].

Spinal cord morphometry will be performed on spine T2 images using the Spinal Cord Toolbox (SCT) [75]. Spinal cord DTI data will be analysed using FSL and SCT. Spine MRS data will be analysed using the linear combination of model spectra (LCModel) [76, 77].

Each of these analysis methods will be considered independently. The *p* values will be corrected for type 1 error inflation using Stepdown Bonferroni testing within each modality.

Statistical analysis.

**Analysis of cross-sectional differences at baseline.** Individuals with FRDA and control groups will be compared at baseline using adjusted and unadjusted analyses. For unadjusted analyses, the two groups will be compared using the two-sample Student's t-test or using the Mann-Whitney U-test for those variables that exhibit substantial skewness or otherwise do not match the assumptions of the two-sample t-test. For adjusted analysis a linear modelling approach will be used to predict case/control status, with demographic variables and site effects as covariates. This analysis will be conducted across all age groups represented in the study design, further enabling the possibility to observe differences in biomarkers between children and adults with FRDA and matched control participants. The baseline analysis provides the starting point from which functions such as rate of change can be derived for each of the parameters.

**Adjustment for potential site differences.** Differences between sites will be examined at baseline, for each primary outcome using a mixed linear model with a random effect of site and fixed effects of age, sex, education, disease severity and other important covariates across individuals with FRDA and control participants and for control participants only. If site differences are detected, we will include site as a random effect within a mixed modelling approach for analyses involving that outcome. This is an important aspect in ensuring robustness in clinical trial ready neuroimaging biomarkers. Furthermore, for a biomarker to be clinical trial ready imaging measures must be reproducible, robust to scanner differences and sensitive to disease-related change. Adhering to these criteria allows the biomarkers to be used in multiple clinical trial contexts, e.g.: defining a clinical trial population Examining differences in data between sites will allow for this distinction to be made and internally validate the measures and identify biomarkers that are most sensitive to change over time in those in the earlier stages of FRDA.

**Assessing the association between variables at baseline.** The associations between variables will be assessed at baseline. For outcomes that have nominally significant differences

between FRDA and control groups at baseline, we will assess their relationship with the number of GAA repeats on both alleles, age of onset and disease duration using Pearson or Spearman's rank correlation. For instance, this analysis can be used to determine whether a monotonic relationship exists between a biomarker and disease severity. We will further assess the correlations between imaging and blood biomarkers with cognitive assessments for participants using a linear modelling framework.

**Assessing the longitudinal changes over time.** The within-person longitudinal slopes across the multiple visits will be estimated using a mixed linear model in which the slope for each participant is treated as a random effect. Within this framework we will assess the significance of a fixed effect that captures the difference in mean slope for individuals with FRDA with respect to controls. Due to the participant demographic in this study the analysis of within person slopes can give an indication of how age of onset and disease stage/severity can influence the subsequent clinical changes that occur within the parameters of participant demography.

**Difference-in-difference analyses.** Using baseline and 12-month follow-up data, we will determine whether the changeover 12-months differs between control and FRDA participants for each outcome. The difference between 12 months and baseline will be determined for each individual, and these differences will be compared between the two groups using a two-sample t-test or Mann-Whitney U-test.

**Stratified analysis.** Additional analyses on data stratified by variables such as disease duration, age of onset and GAA repeat length will be conducted. This stratified analysis will determine whether sensitivity of a biomarker differs across subgroups of FRDA, e.g.: a specific measure has greater sensitivity to change in individuals with an early onset of disease compared with those who have later onset of disease.

**Interim analyses and accounting for missing data.** Interim analysis will be performed after the completion of each visit (baseline, 12-month and 24-month time points). There are likely to be instances of missing data in this study, due to the study requiring interstate travel to attend testing, participant health concerns and normal participant dropout. We had increased our target sample size by 25% to account for 20% anticipated participant attrition, and if required the consortium will consider the recruitment of additional participants in order to ensure that the analysis has the necessary statistical power. Participants with missing data for just one of the three visits will be included under a missing at random assumption for longitudinal mixed linear modelling.

*Data confidentiality and storage*. Participants' personal information, genetic information, clinical, cognitive, mood and biospecimen data generated in this study will be de-identified by assigning a unique study identification number. All raw, pre-processed (defaced) and processed MRI images will be stored in the Monash XNAT platform [78]. Defacing of 3D T1 and T2 MRI data will be conducted using MaskFace [79] version 10.15.18 [80]. To ensure consistency, the team at the University of Minnesota provided each site with a Docker or Singularity container containing scripts to automatically sort the data and deface the brain T1 and T2 images before upload to XNAT [78]. The REDCap database (Helix, Monash University) [81] will be used to store all other study data. All paper-based case report forms (CRFs) will be kept in a locked cabinet to be accessed by authorised personnel at each site according to the Good Clinical Practice guidelines. Data entry into REDCap will follow standard double data entry processes, with responsibility for accuracy lying with individual sites. The central study coordinator will provide overall oversight of data management through randomised checks of a proportion of REDCap entries.

Academic sites and industry collaborators will have secure access to both raw and processed data within the Monash XNAT and REDCap platforms throughout the duration of the study

in accordance with the study agreements. Data may also be made available to third parties at the completion of the study, and upon request, provided that they obtain approval from the TRACK-FA Steering Committee. All records and samples will be stored according to local IRB requirements at each site.

*Quality management, assurance, and control.* Rigorous pilot testing of n = 5 control participants, at each study site, was undertaken prior to study commencement. All pilot images were examined by the Minnesota University imaging team prior to commencement of the study to verify the quality of MRI data.

IXICO has been appointed as a contract research organisation (CRO) through a fee for service contract. IXICO will perform independent quality control and data analysis of brain anatomical imaging and brain diffusion imaging data for one third of TRACK-FA subjects. 300 datasets will be sampled by a TRACK-FA biostatistician who will not be provided the unique study IDs of the selected datasets. Specific analyses performed by IXICO will include volumetric MRI of the cerebellum and brain stem and DTI for cerebellar peduncles, using IXICO's own software and pipelines, which are different from pipelines used by academic sites for primary outcome measures. IXICO will conduct interim analyses at 50% of total enrolment and 100% of total enrolment at each time point. Quality control of MRI procedures will be upheld by conducting phantom quality assurance testing. This testing will be conducted by each study site monthly and will be uploaded to XNAT upon which time any discrepancies will be notified, and modifications will be made to the process at that specific site.

Furthermore, rigorous training has been provided across all study sites to avoid any potential bias in clinical, cognitive and mood test administration. Videos have also been recorded for training purposes and monthly coordinator meetings ensure that study sites have an opportunity to trouble shoot any issues at the local level.

## Status and timeline of the study

TRACK-FA participant recruitment will occur throughout the period of January 2021 to October 2022. As of September 2021, total of a total of 27 individuals with FRDA consisting of 21 adult participants with FRDA, five older paediatric participants with FRDA and 1 younger paediatric participant with FRDA, have been tested. A total of 11 control participants have been tested.

The target recruitment statistics for the 18-month period will include a total of 200 participants with FRDA, including 110 adult participants with FRDA, 82 participants with FRDA of age 11–17 and eight total participants of age 5–10 (Fig 2).

**COVID-19 impact.** The TRACK-FA study has been hindered in its progression due to the COVID-19 global pandemic. Minimising the risk of exposure to COVID-19 for individuals with FRDA has created challenges in a study that depends on face-to-face contact. Site-specific lockdowns, differing vaccination uptake and COVID-related reluctance to travel have also severely limited recruitment. To date, the recruitment targets remain unchanged, and the study is proceeding with a six-month extension for enrolment and assessments as part of visit 1.

## Discussion

There is an urgent need to develop biomarkers for FRDA that can help in the design and monitoring of clinical trials. Specific anatomic targets in the brain and the spinal cord may be more or less relevant at different stages of disease progression while biomarkers that can monitor the pharmacodynamic effects and disease progression are needed as outcome assessments [15, 38, 82]. Neuroimaging outcome measures show great promise in this area, however further work

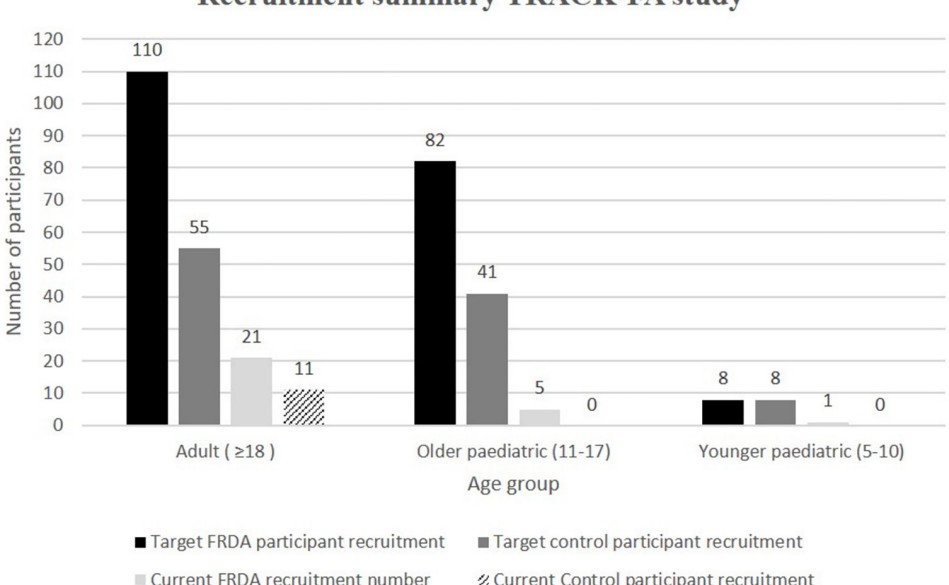

**Fig 2. Summary of recruitment statistics across seven international study sites for the TRACK-FA study.** The target recruitment and the cumulative total of FRDA and control participants currently recruited into the study (categorised into the age groups) have been indicated.

is needed to fully understand the relationship between the neuroimaging outcome measure and disease progression. TRACK-FA is a multimodal, multicentre, longitudinal study which aims to accelerate the discovery of clinical-trial ready biomarkers of disease progression to support this unmet need.

The multicentre nature of the study allows a wider and larger demographic of individuals with FRDA with varying disease severities to be captured, overcoming one of the biggest hindrances faced in biomarker research in rare diseases. Importantly, this large cohort will facilitate targeted subgroup analysis of sufficient statistical power to tease apart the relationship between the underlying neuropathology, and parameters such as clinical presentation, age of onset, age, and genotype. This information will not only guide the validation of neuroimaging biomarkers providing a pre-harmonised set of FRDA imaging centres, which can be used in future trials, but also inform drug development strategy by identifying regions that should be targeted by interventions for maximum impact within a given cohort.

TRACK-FA uses a combination of harmonised MRI modalities to create a unique imaging protocol. The multi-modal combination of brain and spinal cord structural MRI, QSM, diffusion MRI and MRS will allow us to determine the extent, timing and effect size of changes that occur in the brain and the spinal cord with each modality disease progression with higher precision [37]. Moreover, the harmonisation of these modalities will allow for validation of results across different sites and scanners. This will lead to an increased robustness of the biomarkers and their pertinence to detect real change in a potential therapeutics pharmacodynamic effect and/or disease progression, adding to their "clinical trial readiness". These chosen outcome measures have shown great promise to monitor the effects of neuronal damage and gliosis associated with neurodegeneration [83] and provide an opportunity to obtain an holistic image of disease progression at multiple levels of the brain and spinal cord. For example, in the brain the outcome measures will include iron levels within the dentate nuclei [40], volumetric changes in a range of brain substructures [15, 16], and microstructural alterations in the

orientation and organisation of white matter fibres [83]. In the spinal cord, the outcome measures will include the cross-sectional area of the cervical spinal cord, diffusion parameters in the white matter [15, 39] and the ratio of key metabolites, such as N-acetyl aspartate and myo-inositol [83].

In addition to neuroimaging measures, TRACK-FA will collect clinical, cognitive, mood, speech and blood biomarker measures of disease progression. Correlation between these outcome measures will be used to understand the functional utility of specific neuroimaging biomarkers for tracking disease progression and their clinical relevance. This data can also be used individually to evaluate disease associated temporal changes and measure the clinical efficacy of current and future therapeutics aimed towards the improvement of cognition and mood as well as dysarthria in FRDA. Furthermore, the proposed subgroup analysis, allows us to determine how factors such as age of onset and GAA repeat size, correlate to neuroimaging measures, and how each of these factors contribute to FRDA neuropathology [24, 84] contributing to the possible development of therapeutic targets specific for age of onset.

TRACK-FA will accelerate the drug discovery pipeline in a climate where urgency is required for clinical trial-ready biomarkers. A clinical trial-ready biomarker must be reproducible and robust to changes in variables [85, 86]. In this study validation of biomarkers is achieved through several different methods, including a multicentre approach, independent imaging analysis performed by IXICO, control of MRI scanner differences, and the relationship of neuroimaging measures to functional and biospecimen measures. This use of a harmonized and multicentre approach aims to deliver a large multi-modal neuroimaging dataset which in the future can be used to assess the variability and generalizability of finding across studies, more robustly inform future clinical trial design and serve to combine with prospectively acquired interventional data enhancing the statistical power of future studies [87, 88].

## Conclusion

TRACK-FA brings together academic researchers, in collaboration with industry partners and advocacy groups, to accelerate the drug discovery effort in FRDA. Measurements of brain and spinal cord structure, connectivity, and neurochemistry, obtained using harmonised and multi-modal neuroimaging methods, will offer cutting-edge outcome measures that can more directly assess neuropathology and disease trajectory and a set of neuroimaging measures that are capable of evaluating pharmacodynamic and pharmacokinetic properties of current and future therapeutics. The broad age and disease severity range will ensure the mapping of different trajectories of disease progression that may enable more targeted treatments in forestalling disease progression for different FRDA cohorts.

## Acknowledgments

The authors gratefully thank all of the participants for their participation in the study, the Friedreich's Ataxia Research Alliance, Takeda Pharmaceuticals Company, Novartis Gene Therapies, IXICO plc and PTC Therapeutics for their help with FRDA participant recruitment.

We thank Dr Andreas Deistung for providing the sequence parameters for brain QSM.

Study data were collected and managed using REDCap electronic data capture tools hosted and managed by Helix (Monash University).

Study data were collected, stored, and managed using XNAT, a software framework for managing neuroimaging laboratory data hosted by Monash (MXNAT) and Monash Biomedical Imaging (MBI-XNAT).

The authors thank the research coordinators across all TRACK-FA sites for their assistance in participant recruitment and testing.

## Author Contributions

**Conceptualization:** Nellie Georgiou-Karistianis, Louise A. Corben, Kathrin Reetz, Imis Dogan, Jennifer Farmer, Marcondes C. França, Ian H. Harding, Steven Hersch, Michelle L. Krishnan, Thiago J. R. Rezende, Sandro Romanzetti, Christophe Lenglet, Pierre-Gilles Henry.

**Data curation:** Isaac M. Adanyeguh, Ian H. Harding, Christophe Lenglet, Pierre-Gilles Henry.

**Formal analysis:** Richard Joules, Michelle Lax.

**Funding acquisition:** Nellie Georgiou-Karistianis, Louise A. Corben, Kathrin Reetz, Imis Dogan, Jennifer Farmer, Marcondes C. França, Ian H. Harding, Thiago J. R. Rezende, Sandro Romanzetti, Christophe Lenglet, Pierre-Gilles Henry.

**Investigation:** Nellie Georgiou-Karistianis, Louise A. Corben, Kathrin Reetz, Manuela Corti, Martin B. Delatycki, Imis Dogan, Jennifer Farmer, Marcondes C. França, William Gaetz, Ian H. Harding, David Lynch, Thomas Mareci, Sahan Muthuhetti Gamage, Thiago J. R. Rezende, Timothy P. L. Roberts, Jens T. Rosenberg, Sandro Romanzetti, Jörg B. Schulz, Sub Subramony, Stephen Zicha, Christophe Lenglet, Pierre-Gilles Henry.

**Methodology:** Nellie Georgiou-Karistianis, Louise A. Corben, Kathrin Reetz, Isaac M. Adanyeguh, Manuela Corti, Dinesh K. Deelchand, Martin B. Delatycki, Imis Dogan, Rebecca Evans, Jennifer Farmer, Marcondes C. França, William Gaetz, Ian H. Harding, Steven Hersch, Richard Joules, James J. Joers, Michelle L. Krishnan, Michelle Lax, Eric F. Lock, David Lynch, Thomas Mareci, Marina Papoutsi, Thiago J. R. Rezende, Timothy P. L. Roberts, Jens T. Rosenberg, Sandro Romanzetti, Jörg B. Schulz, Adam J. Schwarz, Sub Subramony, Stephen Zicha, Christophe Lenglet, Pierre-Gilles Henry.

**Project administration:** Nellie Georgiou-Karistianis, Louise A. Corben, Kathrin Reetz, Manuela Corti, Martin B. Delatycki, Imis Dogan, Jennifer Farmer, Marcondes C. França, William Gaetz, Ian H. Harding, Karen S. Harris, David Lynch, Thomas Mareci, Thiago J. R. Rezende, Timothy P. L. Roberts, Sandro Romanzetti, Jörg B. Schulz, Sub Subramony, Stephen Zicha, Christophe Lenglet, Pierre-Gilles Henry.

**Software:** Isaac M. Adanyeguh, Thiago J. R. Rezende.

**Validation:** James J. Joers, Eric F. Lock, Marina Papoutsi.

**Visualization:** Ian H. Harding, Christophe Lenglet, Pierre-Gilles Henry.

**Writing – review & editing:** Nellie Georgiou-Karistianis, Louise A. Corben, Kathrin Reetz, Isaac M. Adanyeguh, Manuela Corti, Dinesh K. Deelchand, Martin B. Delatycki, Imis Dogan, Rebecca Evans, Jennifer Farmer, Marcondes C. França, William Gaetz, Ian H. Harding, Karen S. Harris, Steven Hersch, Richard Joules, James J. Joers, Michelle L. Krishnan, Michelle Lax, Eric F. Lock, David Lynch, Thomas Mareci, Sahan Muthuhetti Gamage, Massimo Pandolfo, Marina Papoutsi, Thiago J. R. Rezende, Timothy P. L. Roberts, Sandro Romanzetti, Jörg B. Schulz, Traci Schilling, Adam J. Schwarz, Sub Subramony, Bert Yao, Stephen Zicha, Christophe Lenglet, Pierre-Gilles Henry.

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
