## [Decision Letter · Decision Letter 0]

18 Mar 2022

PONE-D-21-38765A natural history study to track brain and spinal cord changes in individuals with Friedreich's ataxia: TRACK-FA study protocolPLOS ONE

Dear Dr. Georgiou-Karistianis,

Thank you for submitting your manuscript to PLOS ONE. After careful consideration, we feel that it has merit but does not fully meet PLOS ONE’s publication criteria as it currently stands. Therefore, we invite you to submit a revised version of the manuscript that addresses the points raised during the review process. I apologize for the length of time it has taken to render a decision. In the end, I only received one review, but based on my own reading of the manuscript, I agree that only minor revisions are required.

We look forward to receiving your revised manuscript.

Kind regards,

Niels Bergsland

Academic Editor

PLOS ONE

Journal Requirements:

Reviewers' comments:

Reviewer's Responses to Questions

**Comments to the Author**

1. Does the manuscript provide a valid rationale for the proposed study, with clearly identified and justified research questions?

Reviewer #1: Yes

2. Is the protocol technically sound and planned in a manner that will lead to a meaningful outcome and allow testing the stated hypotheses?

Reviewer #1: Yes

3. Is the methodology feasible and described in sufficient detail to allow the work to be replicable?

Reviewer #1: Yes

4. Have the authors described where all data underlying the findings will be made available when the study is complete?

Reviewer #1: Yes

5. Is the manuscript presented in an intelligible fashion and written in standard English?

Reviewer #1: Yes

6. Review Comments to the Author

You may also provide optional suggestions and comments to authors that they might find helpful in planning their study.

Reviewer #1: This is a well planned study that will help to adress the need for Imaging biomarkers to detect sooner improvement/deterioration of the FRDA evolution.

My feeling is that the authors should maybe detail how this study fits in their different Imaging projects in FRDA (Image, Enigma,...) and what it would add compared to previous work, including their previous longitudinal follow-up published in 2021 (obviously much higher number of subjects) and may be summurize the main findings of their other projects that helped design this study.

It would help contextualize the project.

Otherwise it is a work, i look forxard to see the results

7. PLOS authors have the option to publish the peer review history of their article (what does this mean?). If published, this will include your full peer review and any attached files.

Reviewer #1: No

---

## [Author Response · Author response to Decision Letter 0]

4 May 2022

Thank you for your comments. Please see below for a summary of our responses and changes. 

Responses to Issues:

Issue 1: Your ethics statement should only appear in the Methods section of your manuscript. If your ethics statement is written in any section besides the Methods, please move it to the Methods section and delete it from any other section. Please ensure that your ethics statement is included in your manuscript, as the ethics statement entered into the online submission form will not be published alongside your manuscript.

Authors’ revisions in response to Issue 1. Our Ethics Statement is in the Methods section; we have added additional detail to the Ethics Statement, and have moved some details about the consenting process from other sections of the manuscript to the Ethics Statement section, as follows: 

Ethics Statement – Page 12 (of version without tracked changes)

“Potential participants will be provided with information about the study via an IRB-approved letter and information sheet. Research personnel will explain all study procedures to potential participants. Participants will be enrolled into the study upon obtaining written informed consent and meeting the inclusion and exclusion criteria (see below). Opt-in and opt-out systems are in place for all participants. All participant data will be de-identified upon enrolment into the study using a unique participant identifier. Each site will be required to ensure that participants are consented in such a way that allows the sharing of de-identified data (as outlined in below).” 

Issue 2: We note that you have indicated that data from this study may be available upon request. PLOS only allows data to be available upon request if there are legal or ethical restrictions on sharing data publicly. Although all articles must include a Data Availability Statement, some submissions, such as Registered Report Protocols and Lab or Study Protocol articles, may not contain data. For manuscripts that do not report data, authors must state in their Data Availability Statement that their article does not report data and the data availability policy is not applicable to their article.

Authors’ revisions in response to Issue 2. We have clarified that, as a study protocol, our article does not report data; we have moved some text that was previously in the Data Availability Statement to the Data Sharing section of the Methods section, as follows: 

Data Availability – Page 5 

“This article is a study protocol and does not report data; therefore, the data availability policy is not applicable.”

Data sharing – Page 12 (of version without tracked changes)

“All parties will have full access to the de-identified raw data throughout the study and interim reporting and final reports will be provided throughout the study in accordance with the milestones and deliverables. 

All academic partners will provide access to their data to the TRACK-FA Neuroimaging Consortium members and the study funders, as well as the right to create derivative works of the research data.”

We hope you find our revisions satisfactory and look forward to hearing from you. 

Sincerely, 

Professor Nellie Georgiou-Karistianis

---

## [Editor Report · Decision Letter 1]

26 May 2022

A natural history study to track brain and spinal cord changes in individuals with Friedreich's ataxia: TRACK-FA study protocol

PONE-D-21-38765R1

Dear Dr. Georgiou-Karistianis,

We’re pleased to inform you that your manuscript has been judged scientifically suitable for publication and will be formally accepted for publication once it meets all outstanding technical requirements.

Kind regards,

Niels Bergsland

Academic Editor

PLOS ONE
---

## [Editor Report · Acceptance letter]

2 Jun 2022

PONE-D-21-38765R1 

A natural history study to track brain and spinal cord changes in individuals with Friedreich's ataxia: TRACK-FA study protocol 

Dear Dr. Georgiou-Karistianis:

I'm pleased to inform you that your manuscript has been deemed suitable for publication in PLOS ONE. Congratulations! Your manuscript is now with our production department. 

Kind regards, 

on behalf of

Dr. Niels Bergsland 

Academic Editor

PLOS ONE